# Multivariate tests of association based on univariate tests

**Ruth Heller**
Department of Statistics and Operations Research
Tel-Aviv University
Tel-Aviv, Israel 6997801
ruheller@gmail.com

**Yair Heller**
heller.yair@gmail.com

## Abstract

For testing two vector random variables for independence, we propose testing whether the distance of one vector from an arbitrary center point is independent from the distance of the other vector from another arbitrary center point by a univariate test. We prove that under minimal assumptions, it is enough to have a consistent univariate independence test on the distances, to guarantee that the power to detect dependence between the random vectors increases to one with sample size. If the univariate test is distribution-free, the multivariate test will also be distribution-free. If we consider multiple center points and aggregate the center-specific univariate tests, the power may be further improved, and the resulting multivariate test may have a distribution-free critical value for specific aggregation methods (if the univariate test is distribution free). We show that certain multivariate tests recently proposed in the literature can be viewed as instances of this general approach. Moreover, we show in experiments that novel tests constructed using our approach can have better power and computational time than competing approaches.

## 1 Introduction

Let $X \in \Re^p$ and $Y \in \Re^q$ be random vectors, where $p$ and $q$ are positive integers. The null hypothesis of independence is $H_0 : F_{XY} = F_X F_Y$, where the joint distribution of $(X, Y)$ is denoted by $F_{XY}$, and the distributions of $X$ and $Y$, respectively, by $F_X$ and $F_Y$. If $X$ is a categorical variable with $K$ categories, then the null hypothesis of independence is the null hypothesis in the $K$-sample problem, $H_0 : F_1 = \ldots = F_K$, where $F_k, k \in \{1, \ldots, K\}$ is the distribution of $Y$ in category $k$.

The problem of testing for independence of random vectors, as well as the $K$-sample problem on a multivariate $Y$, against the general alternative $H_1 : F_{XY} \neq F_X F_Y$, has received increased attention in recent years. The most common approach is based on pairwise distances or similarity measures. See (26), (6), (24), and (12) for consistent tests of independence, and (10), (25), (1), (22), (5), and (8) for recent $K$-sample tests. Earlier tests based on nearest neighbours include (23) and (13). For the $K$-sample problem, the practice of comparing multivariate distributions based on pairwise distances is justified by the fact that, under mild conditions, the distributions differ if and only if the distributions of within and between pairwise distances differ (19). Other innovative approaches have also been considered in recent years. In (4) and (28), the authors suggest to reduce the multivariate data to a lower dimensional sub-space by (random) projections. Recently, in (3) another approach was introduced for the two sample problem, which is based on distances between analytic functions representing each of the distributions. Their novel tests are almost surely consistent when randomly selecting locations or frequencies and are fast to compute.

We suggest the following approach for testing for independence: first compute the distances from a fixed center point, then apply any univariate independence test on the distances. We show that this approach can result in novel powerful multivariate tests, that are attractive due to their theoretical guarantees and computational complexity. Specifically, in Section 2 we show that if $H_0$ is false, then applying a univariate consistent test on distances from a single center point will result in a multivariate consistent test (except for a measure zero set of center points), where a consistent test is a test with power (i.e., probability of rejecting $H_0$ when $H_0$ is false) increasing to one as the sample size increases when $H_0$ is false. Moreover, the computational time is that of the univariate test, which means that it can be very fast. In particular, a desirable requirement is that the null distribution of the test statistic does not depend on the marginal distributions of $X$ and $Y$, i.e., that the test is distribution-free. Powerful univariate consistent distribution-free tests exist (see (11) for novel tests and a review), so if one of these distribution-free univariate test is applied on the distances, the resulting multivariate test is distribution-free.

In Section 3 we show that considering the distances from $M > 1$ points and aggregating the resulting statistics can also result in consistent tests, which may be more powerful than tests that consider a single center point. Both distribution-free and permutation-based tests can be generated, depending on the choice of aggregation method and univariate test.

In Section 4 we draw the connection between these results and some known tests mentioned above. The tests of (10) and of (12) can be viewed as instances of this approach, where the fixed center point is a sample point, and all sample points are considered each in turn as a fixed center point, for a particular univariate test. In Section 5 we demonstrate in simulations that novel tests based on our approach can have both a power advantage and a great computational advantage over existing multivariate tests. In Section 6 we discuss further extensions.

## 2 From multivariate to univariate

We use the following result by (21). Let $B_d(x, r) = \{y \in \Re^d : \|x - y\| \le r\}$ be a ball centered at $x$ with radius $r$. A complex Radon measure $\mu$, defined formally in Supplementary Material (SM) § D, on $\Re^d$ is said to be of at most exponential-quadratic growth if there exist positive constants $A$ and $\alpha$ such that $|\mu|(B_d(0, r)) \le Ae^{\alpha r^2}$.

**Proposition 2.1** (Rawat and Sitaram (21)). *Let $\Gamma \subset \Re^d$ be such that the only real analytic function (defined on an open set containing $\Gamma$) that vanishes on $\Gamma$, is the zero function. Let $\mathcal{C} = \{B_d(x, r) : x \in \Gamma, r > 0\}$. Then for any complex Radon measure $\mu$ on $\Re^d$ of at most exponential-quadratic growth, if $\mu(C) = 0$ for all $C \in \mathcal{C}$, then it necessarily follows that $\mu = 0$.*

For the two-sample problem, let $Y \in \Re^q$ be a random variable with cumulative distribution $F_1$ in category $X = 1$, and $F_2$ in category $X = 2$. For $z \in \Re^q$, let $F'_{iz}$ be the cumulative distribution function of $\|Y - z\|$ when $Y$ has cumulative distribution $F_i$, $i \in \{1, 2\}$. We show that if the distribution of $Y$ differs across categories, then so does the distribution of the distance of $Y$ from almost every point $z$. Therefore, any univariate consistent two-sample test on the distances from $z$ results in a consistent test of the equality of the multivariate distributions $F_1$ and $F_2$, for almost every $z$. It is straightforward to generalize these results to $K > 2$ categories.

Proofs of all Theorems are in SM § A.

**Theorem 2.1.** *If $H_0 : F_1 = F_2$ is false, then for every $z \in \Re^q$, apart from at most a set of Lebesgue measure 0, there exists an $r > 0$ such that $F'_{1z}(r) \ne F'_{2z}(r)$.*

**Corollary 2.1.** *For every $z \in \Re^q$, apart from at most a set of Lebesgue measure 0, a consistent two-sample univariate test of the null hypothesis $H'_0 : F'_{1z} = F'_{2z}$ will result in a multivariate consistent test of the null hypothesis $H_0 : F_1 = F_2$.*

For the multivariate independence test, let $X \in R^p$ and $Y \in \Re^q$ be two random vectors with marginal distributions $F_X$ and $F_Y$, respectively, and with joint distribution $F_{XY}$. For $z = (z_x, z_y), z_x \in \Re^p, z_y \in \Re^q$, let $F'_{XYz}$ be the joint distribution of $(\|X - z_x\|, \|Y - z_y\|)$. Let $F'_{Xz}$ and $F'_{Yz}$ be the marginal distribution of $\|X - z_x\|$ and $\|Y - z_y\|$, respectively.

**Theorem 2.2.** *If $H_0 : F_{XY} = F_X F_Y$ is false, then for every $z_x \in \Re^p, z_y \in \Re^q$, apart from at most a set of Lebesgue measure 0, there exists $r_x > 0, r_y > 0$, such that $F'_{XYz}(r_x, r_y) \ne F'_{Xz}(r_x)F'_{Yz}(r_y)$.*

**Corollary 2.2.** *For every $z \in \Re^{p+q}$, apart from at most a set of Lebesgue measure 0, a consistent univariate test of independence of the null hypothesis $H'_0 : F'_{XYz} = F'_{Xz}F'_{Yz}$ will result in a multivariate consistent test of the null hypothesis $H_0 : F_{XY} = F_X F_Y$.*

We have $N$ independent copies $(x_i, y_i)$ $(i = 1, \ldots, N)$ from the joint distribution $F_{XY}$. The above results motivate the following two-step procedure for the multivariate tests. For the $K$-sample test, $x_i \in \{1, \ldots, K\}$ determines the category and $y_i \in \Re^q$ is the observation in category $x_i$, so the two-step procedure is to first choose $z \in \Re^q$ and then to apply a univariate $K$-sample consistent test on $(x_1, \|y_1 - z\|), \ldots, (x_N, \|y_N - z\|)$. Examples of such univariate tests include the classic Kolmogorov-Smirnov and Cramer-von Mises tests. For the independence test, the two-step procedure is to first choose $z_x \in \Re^p$ and $z_y \in \Re^q$, and then to apply a univariate consistent independence test on $(\|x_1 - z_x\|, \|y_1 - z_y\|), \ldots, (\|x_N - z_x\|, \|y_N - z_y\|)$. An example of such a univariate test is the classic test of Hoeffding (14). Note that the consistency of a univariate test may be satisfied only under some assumptions on the distribution of the distances of the multivariate vectors. For example, the consistency of (14) follows if the densities of $\|X - z_x\|$ and $\|Y - z_y\|$ are continuous. See (11) for additional distribution-free univariate $K$-sample and independence tests.

A great advantage of this two-step procedure is the fact that it has the same computational complexity as the univariate test. For example, if one chooses to use Hoeffding's univariate independence test (14) , then the total complexity is only $O(N \log N)$, which is the cost of computing the test statistic. The $p$-value can be extracted from a look-up table since Hoeffding's test is distribution-free. In comparison, the computational complexity of the multivariate permutation tests of (26) and (12) is $O(BN^2)$, and $O(BN^2 \log N)$, respectively, where $B$ is the number of permutations. For many univariate tests the asymptotic null distribution is known, thus it can be used to compute the significance efficiently without resorting to permutations, which are typically required for assessing the multivariate significance.

Another advantage of the two-step procedure is the fact that the test statistic may be estimating an easily interpretable population value. The univariate test statistics often converge to easily interpretable population values, which are often between 0 and 1. These values carry over to provide meaning to the new multivariate statistics, see examples in equations (1) and (2).

In practice, the choice of the center value from which the distances are measured can have a significant impact on power, as demonstrated in the following example. Let $\sum_{i=1}^k p_i N_2(\mu_i, diag(\sigma_{i1}^2, \sigma_{i2}^2))$ denote the mixture distribution of $k$ bivariate normals, with mean $\mu_i$ and a diagonal covariance matrix with diagonal entries $\sigma_{i1}^2$ and $\sigma_{i2}^2$, denoted by $diag(\sigma_{i1}^2, \sigma_{i2}^2)$, $i = 1, \ldots, k$. Consider the following bivariate two sample problem which is depicted in Figure 1, where $F_1 = \frac{1}{2}N_2(0, diag(1,9)) + \frac{1}{2}N_2(0, diag(100,100))$ and $F_2 = \frac{1}{2}N_2(0, diag(9,1)) + \frac{1}{2}N_2(0, diag(100,100))$. Clearly $F'_{1z}$ has the same distribution as $F'_{2z}$ if $z \in \{(y_1, y_2) : y_1 = y_2 \text{ or } y_1 = -y_2\}$, see Figure 1 (c). In agreement with theorem 2.1 the measure of these non-informative center points is zero. On the other hand, if we use as a center point a point on one of the axes, the distribution of the distances will be very different. See in particular the distribution of distances from the point (0,100) in Figure 1 (b) and the power analysis in Table 2.

## 3 Pooling univariate tests together

We need not rely on a single $z \in \Re^{p+q}$ (or a single $z \in \Re^q$ for the $K$-sample problem). If we apply a consistent univariate test using many points $z_i$ for $i = 1, \ldots, M$ as our center points, where the test is applied on the distances of the $N$ sample points from the center point, we obtain $M$ test-statistics and corresponding $p$-values, $p_1, \ldots, p_M$.

We can use the $p$-values or the test statistics of the univariate tests to design consistent multivariate tests. We suggest three useful approaches. The first approach is to combine the $p$-values, using a combining function $f : [0, 1]^M \to [0, 1]$. Common combining functions include $f(p_1, \ldots, p_M) = \min_{i=1,\ldots,M} p_i$, and $f(p_1, \ldots, p_M) = -2 \sum_{i=1}^M \log p_i$.

The second approach is to combine the univariate test statistics, by a combining function such as the average, maximum, or minimum statistic. These aggregation methods can result in test statistics which converge to meaningful population values, see equations (1) and (2) below for multivariate tests based on the univariate Kolmogorov-Smirnov two sample test (18). We note that if the univariate

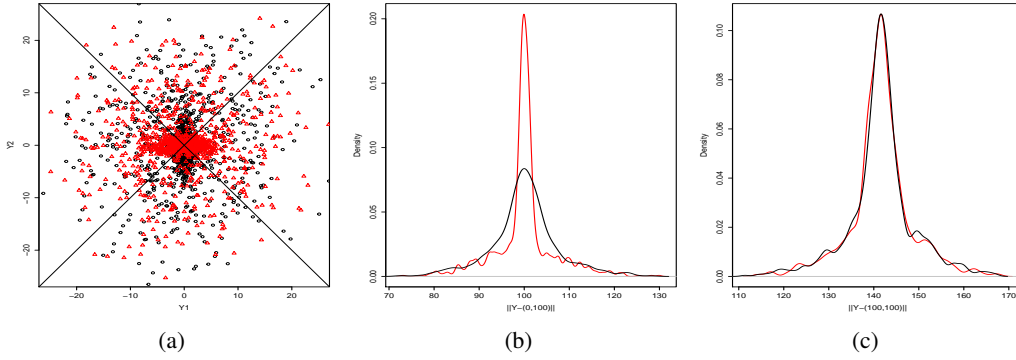

Figure 1: (a) Realizations from two bivariate normal distributions, with a sample size of 1000 from each group: $\frac{1}{2}N_2(0, diag(1, 9)) + \frac{1}{2}N_2(0, diag(100, 100))$ (black points), and $F_2 = \frac{1}{2}N_2(0, diag(9, 1)) + \frac{1}{2}N_2(0, diag(100, 100))$ (red points); (b) the empirical density of the distance from the point $(0,100)$ in each group; (c) the empirical density of the distance from the point $(100,100)$ in each group.

tests are distribution-free then taking the maximum (minimum) $p$-value is equivalent to taking the minimum (maximum) test statistic (when the test rejects for large values of the test statistic). The significance of the combined $p$-value or the combined test statistic can be computed by a permutation test.

A drawback of the two approaches above is that the distribution-free property of the univariate test does not carry over to the multivariate test. In our third approach, we consider the set of $M$ $p$-values as coming from the family of $M$ null hypotheses, and then apply a valid test of the global null hypothesis that all $M$ null hypotheses are true. Let $p_{(1)} \leq \ldots \leq p_{(M)}$ be the sorted $p$-values. The simplest valid test for any type of dependence is the Bonferroni test, which will reject the global null if $Mp_{(1)} \leq \alpha$.

Another valid test is the test of Hommel (16), which rejects if $\min_{j\geq 1}\{M(\sum_{l=1}^{M} 1/l)p_{(j)}/j\} \leq \alpha$. (This test statistic was suggested independently in a multiple testing procedure for false discovery rate control under general dependence in (2).) The third approach is computationally much more efficient than the first two approaches, since no permutation test is required after the computation of the univariate $p$-values, but it may be less powerful. Clearly, if the univariate test is distribution free, the resulting multivariate test has a distribution-free critical value.

As an example we prove that when using the Kolmogorov-Smirnov two sample test as the univariate test, all the pooling methods above result in consistent multivariate two-sample tests. Let $KS(z) = \sup_{d\in\Re} |F'_{1z}(d) - F'_{2z}(d)|$ be the population value of the univariate Kolmogorov-Smirnov two sample test statistic comparing the distribution of the distances. Let $N$ be the total number of independent observations. We assume for simplicity an equal number of observations from $F_1$ and $F_2$.

**Theorem 3.1.** *Let $z_1, \ldots, z_M$ be a sample of center points from an absolutely continuous distribution with probability measure $\nu$, whose support $S$ has a positive Lebesgue measure in $\Re^q$. Let $KS_N(z_i)$ be the empirical value of $KS(z_i)$ with corresponding $p$-value $p_i$, $i = 1, \ldots, M$. Let $p_{(1)} \leq \ldots \leq p_{(M)}$ be the sorted $p$-values. Assume that the distribution functions $F_1$ and $F_2$ are continuous. For $M = o(e^N)$, if $H_0 : F_1 = F_2$ is false, then $\nu$-almost surely, the multivariate test will be consistent for the following level $\alpha$ tests:*

1. *the permutation test using the test statistics $S1 = \max_{i=1,\ldots,M}\{KS_N(z_i)\}$ or $S2 = p_{(1)}$.*

2. *the test based on Bonferroni, which rejects $H_0$ if $Mp_{(1)} \leq \alpha$.*

3. *for $M \log M = o(e^N)$, the test based on Hommel's global null $p$-value, which rejects $H_0$ if $\min_{j=1,\ldots,M}\left\{M(\sum_{l=1}^{M} 1/l)p_{(j)}/j\right\} \leq \alpha$.*

4. *the permutation tests using the statistics $T1 = \sum_{i=1}^{M} KS_N(z_i)$ or $T2 = -2\sum_{i=1}^{M} \log p_i$.*

Arguably, the most natural choice of center points is the sample points themselves. Interestingly, if the univariate test statistic is a U-statistic (15) of order $m$ (defined formally in SM §sup-sec-technical), then the resulting multivariate test statistic is a U-statistic of order $m + 1$, if each sample point acts as a center point, and the univariate test statistics are averaged, as stated in the following Lemma (see SM § A for the proof).

**Lemma 3.1.** *For univariate random variables* $(U, V)$*, let* $T_{N-1}((u_k, v_k), k = 1, \ldots, N-1)$ *be a univariate test statistic based on a random sample of size* $N - 1$ *from the joint distribution of* $(U, V)$*. If* $T_{N-1}$ *is a U-statistic of order* $m$*, then* $S_N = \frac{1}{N}[T\{(\|x_k - x_1\|, \|y_k - y_1\|), k = 2, \ldots, N\} + \ldots + T\{(\|x_k - x_N\|, \|y_k - y_N\|), k = 1, \ldots, N - 1\}]$ *is a U-statistic of order* $m + 1$*.*

The test statistics $S1$ and $T1/M$ converge to meaningful population quantities,

$$\lim_{N,M \to \infty} S_1 = \lim_{M \to \infty} \max_{z_1, \ldots, z_M} KS(z) = \sup_{z \in S} KS(z), \tag{1}$$

$$\lim_{N,M \to \infty} T_1/M = \lim_{M \to \infty} \sum_{i=1}^{M} KS(z_i)/M = E\{KS(Z)\}, \tag{2}$$

where the expectation is over the distribution of the center point $Z$.

## 4 Connection to existing methods

We are aware of two multivariate test statistics of the above-mentioned form: aggregation of the univariate test statistics on the distances from center points. The tests are the two sample test of (10) and the independence test of (12). Both these tests use the second pooling method mentioned above by summing up the univariate test statistics. Furthermore, both these tests use the $N$ sample points as the center points (or $z$'s) and perform a univariate test on the remaining $N - 1$ points. Indeed, (10) recognized that their test can be viewed as summing up univariate Cramer von-Mises tests on the distances from each sample point. We shall show that the test statistic of (12) can be viewed as aggregation by summation of the univariate weighted Hoeffding independence test suggested in (27).

In (12) a permutation test was introduced, using the test statistic $\sum_{i=1}^{N} \sum_{j=1, j \neq i}^{N} S(i, j)$, where $S(i, j)$ is the Pearson test score for the $2 \times 2$ contingency table for the random variables $I(\|X - x_i\| \leq \|x_j - x_i\|)$ and $I(\|Y - y_i\| \leq \|y_j - y_i\|)$, where $I(\cdot)$ is the indicator function. Since $\|X - x_i\|$ and $\|Y - y_i\|$ are univariate random variables, $S(i, j)$ can also be viewed as the test statistic for the independence test between $\|X - x_i\|$ and $\|Y - y_i\|$, based on the $2 \times 2$ contingency table induced by the $2 \times 2$ partition of $\Re^2$ about the point $(\|x_j - x_i\|, \|y_j - y_i\|)$ using the $N - 2$ sample points $(\|x_k - x_i\|, \|y_k - y_i\|), k = 1, \ldots, N, k \neq i, k \neq j$. The statistic that sums the Pearson test statistics over all $2 \times 2$ partitions of $\Re^2$ based on the observations, results in a consistent independence test for univariate random variables (27). The test statistic of (27) on the sample points $(\|x_k - x_i\|, \|y_k - y_i\|), k = 1, \ldots, N, k \neq i$, is therefore $\sum_{j=1, j \neq i}^{N} S(i, j)$. The multivariate test statistic of (12) aggregates by summation the univariate test statistics of (27), where the $i$th univariate test statistic is based on the $N - 1$ distances of $x_k$ from $x_i$, and the $N - 1$ distances of $y_k$ from $y_i$, for $k = 1, \ldots, N, k \neq i$.

Of course, not all known consistent multivariate tests belong to the framework defined above. As an interesting example we discuss the energy test of (25) and (1) for the two-sample problem. Without loss of generality, let $y_1, \ldots, y_{N_1}$ be the observations from $F_1$, and $y_{N_1+1}, \ldots, y_N$ be the observations from $F_2$, $N_2 = N - N_1$. The test statistic $\mathcal{E}$ is equal to

$$\frac{N_1 N_2}{N} \left( \frac{2}{N_1 N_2} \sum_{l=1}^{N_1} \sum_{m=N_1+1}^{N} \|y_l - y_m\| - \frac{1}{N_1^2} \sum_{l=1}^{N_1} \sum_{m=1}^{N_1} \|y_l - y_m\| - \frac{1}{N_2^2} \sum_{l=N_1+1}^{N} \sum_{m=N_1+1}^{N} \|y_l - y_m\| \right),$$

where $\|\cdot\|$ is the Euclidean norm. It is easy to see that $\mathcal{E} = \sum_{i=1}^{N} S_i$, where the univariate score is

$$S_i = \left\{ \frac{1}{N_1} \sum_{m=1}^{N_1} \|y_i - y_m\| - \frac{1}{N_2} \sum_{m=N_1+1}^{N} \|y_i - y_m\| \right\} w(i), \tag{3}$$

and $w(i) = -\frac{N_2}{N}$ if $i \leq N_1$ and $w(i) = \frac{N_1}{N}$ if $i > N_1$, for $i \in \{1, \ldots, N\}$. The statistic $S_i$ is not an omnibus consistent test statistic, since a test based on $S_i$ will have no power to detect difference in distributions with the same expected distance from $y_i$ across groups. However, the energy test is omnibus consistent.

# 5 Experiments

In order to assess the effect of using our novel approach, we carry out experiments. We have three specific aims: (1) to compare the power of using a single center point versus multiple center points; (2) to assess the effect of different univariate tests on the power; and (3) to see how the resulting tests fare against other multivariate tests. For simplicity, we address the two-sample problem, and we do not consider the more computationally intensive pooling approaches one and two, but rather consider only the third approach that results in a distribution-free critical value for the multivariate test.

**Simulation 1: distributions of dimension** $\geq 2$. We examined the distributions depicted in Figure 2. Scenario (a) was chosen to examine the classical setting of discovering differences in multivariate normal distributions. The other scenarios were chosen to discover differences in the distributions when one or both distributions have clusters. These are similar to the settings considered in (9). In addition, we examined the following scenario from (25) in five dimensions: $F_1$ is the multivariate standard normal distribution, and $F_2 = t(5)^{(5)}$ is the multivariate $t$ distribution, where each of the independent 5 coordinates has the univariate $t$ distribution with five degrees of freedom.

Regarding the choice of center points, we examine as single center point a sample point selected at random or the center of mass (CM), and as multiple center points all sample points pooled by the third approach (using either Bonferroni's test or Hommel's test). Regarding the univariate tests, we examine: the test of Kolmogorov-Smirnov (18), referred to as KS; the test of the Anderson and Darling family, constructed by (20) for the univariate two-sample problem, referred to as AD; the generalized test of (11), that aggregates over all partition sizes using the minimum $p$-value statistic, referred to as $minP$ (see SM § C for a detailed description). We compare our tests to Hotelling's $T^2$ classical generalization of the Student's $t$ statistic for multivariate normal data (17), referred to as Hotelling; to the energy test of (25) and (1), referred to as Edist; and to the maximum mean discrepancy test of (8), referred to as MMD.

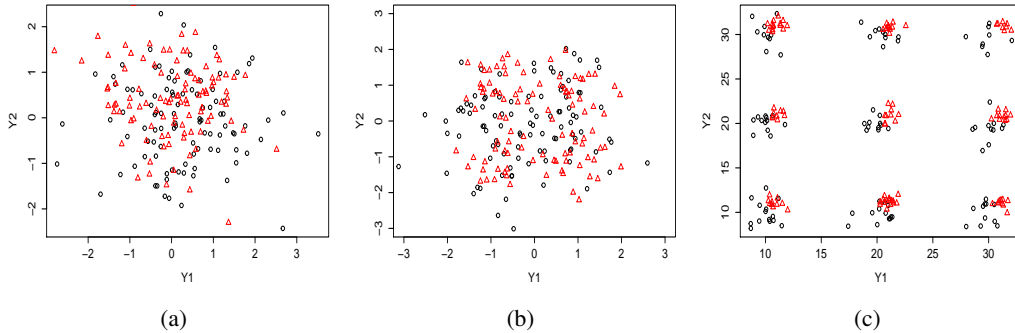

(a)           (b)           (c)

Figure 2: Realizations from the three non-null bivariate settings considered, with a sample size of 100 from each group: (a) $F_1 = N_2\{(0,0), diag(1,1)\}$ and $F_2 = N_2\{(0,0.05), diag(0.9,0.9)\}$; (b) $F_1 = N_2\{(0,0), diag(1,1)\}$ and $F_2 = \sum_{i=1}^{4} \frac{1}{4} N_2\{\mu_i, diag(0.25, 0.25)\}$, where $\mu_1 = c(1,1)$, $\mu_2 = c(-1,1)$, $\mu_3 = c(1,-1)$, $\mu_4 = c(-1,-1)$ ; (c) $F_1 = \sum_{i=1}^{9} \frac{1}{9} N_2\{\mu_i, diag(1,1)\}$ and $F_2 = \sum_{i=1}^{9} \frac{1}{9} N_2\{\mu_i + (1,1), diag(0.25, 0.25)\}$ are both mixtures of nine bivariate normals with equal probability of being sampled, but the centers of the bivariate normals of $F_1$ are on the grid points $(10, 20, 30) \times (10, 20, 30)$ and have covariance $diag(1,1)$, and the centers of the bivariate normals of $F_2$ are on the grid points $(11, 21, 31) \times (11, 21, 31)$ and have covariance $diag(0.25, 0.25)$.

Table 1 shows the actual significance level (column 3) and power (columns 4–7), for the different multivariate tests considered, at the $\alpha = 0.1$ significance level. We see that the choice of center point matters: comparing rows 4–6 to rows 7–9 shows that depending on the data generation, there can be more or less power to the test that selects as the center point a sample point at random, versus the center of mass, depending on whether the distances from the center of mass are more informative than the distances from a random point. Comparing these rows with rows 10–15 shows that in most settings there was benefit in considering all sample points as center points versus only a single center point, even at the price of paying for multiplicity of the different center points. This was true despite

Table 1: The fraction of rejections at the 0.1 significance level for the null case (column 3), the three scenarios depicted in Figure 2 (columns 4–6), and the additional scenario of higher dimension (column 7). The sample size in each group was 100. Rows 4–6 use the center of mass (CM) as a single center point; rows 7–9 use a random sample point as the single center point; rows 10–12 use all sample points as center points. The adjustment for the multiple center points is by Bonferroni in rows 10–12, and by Hommel's test in rows 13–15. Based on 500 repetitions for columns 4–7, and on 1000 repetitions for the true null setting in column 3.

| Row | Test | $F_1 = F_2 =$ $N_2\{(0,0), diag(1,1)\}$ | Scenarios in Figure 2 (a) | (b) | (c) | $N_5\{(0,0), diag(1,1,1,1,1)\}$ $, t(5)^{(5)}$ |
|---|---|---|---|---|---|---|
| 1 | Hotelling | 0.097 | 0.952 | 0.064 | 0.246 | 0.080 |
| 2 | Edist | 0.090 | 0.958 | 0.826 | 0.298 | 0.438 |
| 3 | MMD | 0.114 | 0.908 | 0.926 | 0.190 | 0.682 |
| 4 | single Z-CM - minP | 0.095 | 0.308 | 0.990 | 0.634 | 0.974 |
| 5 | single Z -CM - KS | 0.087 | 0.262 | 0.982 | 0.214 | 0.924 |
| 6 | single Z-CM-AD | 0.112 | 0.350 | 0.994 | 0.266 | 0.978 |
| 7 | single Z -random - minP | 0.097 | 0.504 | 0.736 | 0.922 | 0.754 |
| 8 | single Z -random - KS | 0.099 | 0.502 | 0.702 | 0.394 | 0.656 |
| 9 | single Z - random - AD | 0.102 | 0.556 | 0.708 | 0.436 | 0.750 |
| 10 | vector Z - minP-Bonf | 0.028 | 0.592 | 0.962 | 1.000 | 0.906 |
| 11 | vector Z - KS-Bonf | 0.013 | 0.692 | 0.858 | 0.196 | 0.722 |
| 12 | vector Z-ad-Bonf | 0.011 | 0.772 | 0.820 | 0.132 | 0.778 |
| 13 | vector Z - minP-Hommel | 0.008 | 0.606 | 0.936 | 1.000 | 0.774 |
| 14 | vector Z - KS-Hommel | 0.009 | 0.588 | 0.776 | 0.174 | 0.550 |
| 15 | vector Z-AD-Hommel | 0.007 | 0.720 | 0.760 | 0.150 | 0.668 |

the fact that the cut-off for significance when considering all sample points was conservative, as manifest by the lower significance levels when the null is true (in column 3, rows 10-15 the actual significance level is at most 0.028). Applying Hommel's versus Bonferroni's test matters as well, and the latter has better power in most scenarios. The greatest difference in power is due to the univariate test choice. A comparison of using KS (rows 5, 8, 11, and 14) versus AD (rows 6, 9, 12, and 15) and $minP$ (rows 4, 7, 10, and 13) shows that AD and $minP$ are more powerful than KS, with a large power gain for using $minP$ when there are many clusters in the data (column 6). As expected, Hotelling, Edist and MMD perform best for differences in the Gaussian distribution (column 4). However, in all other settings Hotelling's test has poor power, and our approach with $minP$ as the univariate test has more power than Edist and MMD in columns 5–7. A possible explanation for the power advantage using an omnibus consistent univariate test over Edist is the fact that Edist aggregates over the univariate scores in (3), and the absolute value of these scores is close to zero for sample points that are on average the same distance away from both groups (even if the spread of the distances from these sample points is different across groups), and for certain center points the score can even be negative.

**Simulation 2: a closer inspection of a specific alternative**. For the data generation of Figure 1, we can actually predict which of the partition based univariate tests should be most powerful. This of course requires knowing the data generations mechanism, which is unknown in practice, but it is interesting to examine the magnitude of the gaps in power from using optimal versus other choices of center points and univariate tests. As one intuitively expects, choosing a point on one of the axes gives the best power. Specifically, looking at the densities of the distributions of distances from (0,100) in Figure 1 (b) one can expect that a good way to differentiate between the two densities is by partitioning the sample space into at least five sections, defined by the four intersections of the two densities closest to the center. In the power analysis in Table 2, $M_5$, a test which looks for the best 5-way partition, has the highest power among all $M_k$ scores, $k = 2, 3, \ldots$. Similarly, an $S_k$ score sums up all the scores of partitions into exactly $k$ parts, and we would like a partition to be a refinement of the best five way partition in order for it to get a good score. Here, $S_8$ has the best power among all $S_k$ scores, $k = 2, 3, \ldots$. For more details about these univariate tests see SM § C. In summary, in this specific situation, it is possible to predict both a good center point and a good very specific univariate score. However this is not the typical situation since usually we do not know enough about the alternative and therefore it is best to pool information from multiple center points together as suggested in Section 3, and to use a more general univariate score, such as $minP$, which is the minimum of the $p$-values of the scores $S_k, k \in \{2, 3, \ldots\}$.

We expect pooling methods one and two to be more powerful than the third pooling method used in the current study, since the Bonferroni and Hommel tests are conservative compared to using

Table 2: The fraction of rejections at the 0.1 significance level for testing $H_0 : F_1 = F_2$ when $F_1 = \frac{1}{2}N_2(0, diag(1,9)) + \frac{1}{2}N_2(0, diag(100,100))$ and $F_2 = \frac{1}{2}N_2(0, diag(9,1)) + \frac{1}{2}N_2(0, diag(100,100))$, based on a sample of 100 points from each group, using different univariate tests and different center points schemes. Based on 500 repetitions. The competitors had the following power: Hotelling, 0.090; Edist, 0.274; MMD 0.250.

| Test | Partitions considered | Aggregation type | Single center point | | Sample points are the center points | |
| --- | --- | --- | --- | --- | --- | --- |
| | | | $z = (0, 100)$ | $z = (0, 4)$ | Bonferroni | Hommel |
| minP | all | | 0.896 | 0.864 | 0.870 | 0.758 |
| KS | $2 \times 2$ | maximum | 0.574 | 0.508 | 0.208 | 0.110 |
| AD | $2 \times 2$ | sum | 0.504 | 0.702 | 0.064 | 0.030 |
| $M_5$ | $5 \times 5$ | maximum | 0.850 | 0.834 | 0.904 | 0.644 |
| $S_5$ | $5 \times 5$ | sum | 0.890 | 0.902 | 0.706 | 0.550 |
| $M_8$ | $8 \times 8$ | maximum | 0.820 | 0.794 | 0.856 | 0.586 |
| $S_8$ | $8 \times 8$ | sum | 0.924 | 0.912 | 0.876 | 0.736 |

the exact permutation null distribution of their corresponding test statistics. We learn from the experiments above and in SM § B, that our approach can be useful in designing well-powered tests, but that important choices need to be made, especially the choice of univariate test, for the resulting multivariate test to have good power.

## 6 Discussion

We showed that multivariate $K$-sample and independence tests can be performed by comparing the univariate distributions of the distances from center points, and that favourable properties of the univariate tests can carry over to the multivariate test. Specifically, (1) if the univariate test is consistent then the multivariate test will be consistent (except for a measure zero set of center points); (2) if the univariate test is distribution-free, the multivariate test has a distribution-free critical value if the third pooling method is used; and (3) if the univariate test-statistic is a $U$-statistic of order $m$, then aggregating by summation with the sample points as center points produces a multivariate test-statistic which is a $U$-statistic of order $m + 1$. The last property may be useful in working out the asymptotic null distribution of the multivariate test-statistic, thus avoiding the need for permutations when using the second pooling method. It may also be useful for working out the non-null distribution of the test-statistic, which may converge to a meaningful population quantity.

The experiments show great promise for designing multivariate tests using our approach. Even though only the most conservative distribution-free tests were considered, they had excellent power. The approach is general, and several important decisions have to be made when tailoring a test to a specific application: (1) the number and location of the center points; (2) the univariate test; and (3) the pooling method.We plan to carry out a comprehensive empirical investigation to assess the impact of the different choices. We believe that our approach will generate useful multivariate tests for various modern applications, especially applications where the data are naturally represented by distances such as the study of microbiome diversity (see SM § B for an example).

The main results were stated for given center points, yet in simulations we select the center points using the sample. The theoretical results hold for a center point selected at random from the sample. This can be seen by considering a two-step process, of first selecting the sample point that will be a center point, and then testing the distances from this center point to the remaining N-1 sample points. Since the $N$ sample points are independent, the consistency result holds. However, if the center point is the center of mass, and it converges to a bad point, then such a test will not be consistent. Therefore we always recommend at least one center point randomly sampled from a distribution with a support of positive measure.

Our theoretical results were shown to hold for the Euclidean norm. However, imposing the restriction that the multivariate distribution function is smooth, the theoretical results will hold more generally for any norms or quasi-norms. From a practical point of view, adding a small Gaussian error to the measured signal guarantees that these results will hold for any normed distance.

**Acknowledgments**

We thank Boaz Klartag and Elchanan Mossel for useful discussions of the main results.

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
