[Supplementary Material · paper_supp.pdf]

# Supplementary Material for Multivariate tests of association based on univariate tests

**Ruth Heller**
Department of Statistics and Operations Research
Tel-Aviv University
Tel-Aviv, Israel 6997801
ruheller@gmail.com

**Yair Heller**
heller.yair@gmail.com

## A    Proofs

### A.1    Proof of Theorem 2.1

*Proof.* Suppose by contradiction, that there is a set $\Gamma \subseteq \Re^q$ with positive Lebesgue measure, such that for all $z \in \Gamma$, $F'_{1z}(r) = F'_{2z}(r)$ for all $r > 0$. It follows that $\int_{y \in B_q(z,r)} dF_1(y) - \int_{y \in B_q(z,r)} dF_2(y) = 0$ for all $r > 0$ and $z \in \Gamma$. Since $|F_1 - F_2| \leq 1$, clearly $F_1 - F_2$ is of at most exponential-quadratic growth. Moreover, the only real analytic function that vanishes on $\Gamma$ is the zero function, since $\Gamma$ has positive Lebesgue measure. Therefore, it follows from Proposition 2.1 that $F_1 - F_2 = 0$, thus contradicting the fact that $H_0$ is false. □

### A.2    Proof of Corollary 2.1

*Proof.* If $H_0 : F_1 = F_2$ is false, then Theorem 2.1 guarantees that for every $z$, apart from at most a set of Lebesgue measure zero, the null univariate hypothesis, $H'_0 : F'_{1z} = F'_{2z}$, is false. Since for such a good $z$ the asymptotic power of a false null univariate hypothesis will be one for any consistent two-sample univariate test, the power of the multivariate test will be one. □

### A.3    Proof of Theorem 2.2

*Proof.* Suppose by contradiction, that there is a set $\Gamma \subseteq \Re^{p+q}$ with positive Lebesgue measure, such that for all $z \in \Gamma$, $F'_{XYz}(r_x, r_y) = F'_{Xz}(r_x)F'_{Yz}(r_y)$ for all $r_x > 0, r_y > 0$. It follows that for all $z \in \Gamma$ and $r_x > 0, r_y > 0$,

$$\int_{(\|x-z_x\|, \|y-z_y\|) \leq (r_x, r_y)} dF_{XY}(x,y) = \int_{\|x-z_x\| \leq r_x} dF_X(x) \int_{\|y-z_y\| \leq r_y} dF_Y(y).$$

It thus follows that for all $z \in \Gamma$ and any $r > 0$,

$$\int_{(\|(x,y)-(z_x,z_y)\|) \leq r} dF_{XY}(x,y) = \int_{\|(x,y)-(z_x,z_y)\| \leq r} dF_X(x) dF_Y(y). \tag{1}$$

However, from Theorem 2.1, with $F_1 = F_{XY}$ and $F_2 = F_X F_Y$, it follows that for all $z \in \Gamma$, apart from a set of Lebesgue measure 0, there exists an $r > 0$ such that

$$\int_{(\|(x,y)-(z_x,z_y)\|) \leq r} dF_{XY}(x,y) \neq \int_{\|(x,y)-(z_x,z_y)\| \leq r} dF_X(x) dF_Y(y),$$

thus contradicting (1). □

## A.4 Proof of Corollary 2.2

*Proof.* If $H_0 : F_{XY} = F_X F_Y$ is false, then Theorem 2.2 guarantees that for every $z$, apart from at most a set of Lebesgue measure 0, the null univariate hypothesis, $H_0' : F_{XYz}' = F_{Xz}' F_{Yz}'$, is false. Since for such a $z$ the asymptotic power of a false null univariate hypothesis test will be one for any consistent univariate test of independence, the power of the multivariate test will be one. $\square$

## A.5 Proof of Theorem 3.1

*Proof.* Proving item 2 will prove item 1 since if the Bonferonni adjusted $p$-value is consistent then so is the permutation test based on the minimum p-value (or maximum statistic), which has necessarily a smaller $p$-value than $M p_{(1)}$. We need to show that the probability of rejection goes to one when $H_0$ is false. According to Corollary 2.1 when $H_0$ is false, $\nu$-almost surely any point $z_i$ offers a consistent univariate test. Therefore, $\nu$-almost surely, $\sup_{i=1,\dots,M} KS(z_i) \geq KS(z_1) > 0$. Let $d_0$ be a distance such that $|F_{1z_1}'(d_0) - F_{2z_1}'(d_0)| = c > 0$.

Let $F_{iz_1 N}'$ be the empirical cumulative distribution function based on $N/2$ sampled distances from $F_{iz_1}'$, $i \in \{1, 2\}$. The test statistic is bounded away from zero:

$$\mathrm{pr}\{\sup_{i=1,\dots,M} KS_N(z_i) > c/2)\} \geq \mathrm{pr}\{KS_N(z_1) > c/2\} = \mathrm{pr}\{\sup_d |F_{1z_1 N}'(d) - F_{2z_1 N}'(d)| > c/2\}$$

$$\geq \mathrm{pr}\{|F_{1z_1 N}'(d_0) - F_{2z_1 N}'(d_0)| > c/2\}$$

$$\geq \mathrm{pr}\{|F_{1z_1 N}'(d_0) - F_{1z_1}'(d_0)| < c/4\} \mathrm{pr}\{|F_{2z_1 N}'(d_0) - F_{2z_1}'(d_0)| < c/4\} \geq (1 - 2e^{-Nc^2/8})^2$$

where in the last row, the first inequality follows since if $|F_{1z_1 N}'(d_0) - F_{1z_1}'(d_0)| < c/4$ and $|F_{2z_1 N}'(d_0) - F_{2z_1}'(d_0)| < c/4$, given that $|F_{1z_1}'(d_0) - F_{2z_1}'(d_0)| = c$, it implies that $|F_{1z_1 N}'(d_0) - F_{2z_1 N}'(d_0)| > c/2$, and the last inequality is the Dvoretzky—Kiefer—Wolfowitz inequality (1). Therefore, when $H_0$ is false, the probability that the statistic is greater than $c/2$ goes to 1 as $N \to \infty$.

When $H_0$ is true, let $F_z'$ denote the common cumulative distribution function of $\|Y - z\|$. For each $z \in \{z_1, \dots, z_M\}$,

$$\begin{aligned}
\mathrm{pr}\{KS_N(z) > c/2\} &= \mathrm{pr}\{\sup_d |F_{1zN}'(d) - F_z'(d) + F_z'(d) - F_{2zN}'(d)| > c/2\} \\
&\leq \mathrm{pr}\{\sup_d |F_{1zN}'(d) - F_z'(d)| + \sup_d |F_{2zN}'(d) - F_z'(d)| > c/2\} \\
&\leq \mathrm{pr}\{\sup_d |F_{1zN}'(d) - F_z'(d)| > c/4\} \\
&+ \mathrm{pr}\{\sup_d |F_{2zN}'(d) - F_z'(d)| > c/4\} \leq 4e^{-Nc^2/8},
\end{aligned} \tag{2}$$

where the last inequality follows from the Dvoretzky—Kiefer—Wolfowitz inequality. It follows from (2) that the Bonferonni adjusted p-value is bounded above by $4M e^{-Nc^2/8}$, and therefore goes to zero as $N \to \infty$ for $M = o(e^N)$, proving consistency.

For item 3, the proof is very similar. Hommel's global null $p$-value is at most $M(\sum_{l=1}^{M} 1/l) p_{(1)}$, and as in the proof for item 2 it is bounded above by $4M(\sum_{l=1}^{M} 1/l) e^{-Nc^2/8}$, which goes to zero as $N \to \infty$ for $M \log M = o(e^N)$.

For item 4, let $z_0 \in \Re^q$ be a center point sampled from $\nu$. When $H_0$ is false, $\nu$-almost surely $KS(z_0) = c > 0$. By Lebesgue's density theorem $\nu$-almost surely there exists an $\epsilon$ such that if $r < \epsilon$ then at least half of the ball $B_q(z_0, r)$ is within the support $S$. Since $F_1$ and $F_2$ are continuous, $KS(z)$ is a continuous function of $z$. Therefore, there exists an $\epsilon' < \epsilon$ such that $KS(z) > c/2$ for all $z \in B_q(z_0, \epsilon') \cap S$. Similar arguments to those for item 2 show that $\nu$-almost surely for any $z_i \in S \cap B_q(z_0, \epsilon')$,

$$\mathrm{pr}(KS_N(z_i) < c/4) < 4e^{-Nc^2/32}. \tag{3}$$

Therefore, $\mathrm{pr}(\cup_{z_i \in S \cap B_q(z_0, \epsilon')} KS_N(z_i) < c/4) < 4M e^{-Nc^2/32}$. Since $\nu$-almost surely $\mathrm{pr}\{Z \in S \cap B_q(z_0, \epsilon')\} > 0$, then $\nu$-almost surely with probability going to one $T1$ is $O(M)$, as long as $M = o(e^N)$. On the other hand when $H_0$ is true, $E(KS_N(z)) = O(1/\sqrt{N})$, see for example Marsaglia et al. (7). Therefore, $E(T1) = O(M/\sqrt{N})$, and by Markov's inequality the permutation

test based on $T1$ will have $\nu$-almost surely power increasing to one as the sample size increases. For the test based on $T_2$, from equations (3) and (2) it follows that for $N$ large enough $p_i < 4e^{-Nc^2/32}$ for $z_i \in S \cap B_q(z_0, \epsilon')$, $i = 1, \ldots, M$. Therefore, for $N$ large enough $-2\sum_{i=1}^M \log p_i$ is greater than $O(NM)\mathrm{pr}\{Z \in S \cap B_q(z_0, \epsilon')\}$. On the other hand, when $H_0$ is true $P_i$ is uniformly distributed, so $E(-2\sum_{i=1}^M \log P_i)$ is $O(M)$. By Markov's inequality the permutation test based on $-2\sum_{i=1}^M \log p_i$ will have $\nu$-almost surely power increasing to one as the sample size increases. $\square$

### A.6   Proof of Lemma 3.1

Denote $B(a, b)$ for the binomial coefficient $a$ choose $b$. If the univariate test statistic $T_{N-1}$ is a $U$-statistic, then it can be written as $T_{N-1} = \sum_{C_{N-1,m}} h\{(u_{j_1}, v_{j_1}), \ldots, (u_{j_m}, v_{j_m})\}/B(N-1, m)$, where $h$ is a symmetric function, $(u_{j_1}, v_{j_1}), \ldots, (u_{j_m}, v_{j_m})$ is a subset of size $m$ from a sample of size $N - 1$, and $C_{N-1,m}$ is the set of all such subsets of size $m$. The multivariate test statistic $S_N$ can therefore be written as

$$\sum_{C_{N,m+1}} f\{(x_{j_1}, y_{j_1}), \ldots, (x_{j_{m+1}}, y_{j_{m+1}})\}/B(N, m+1),$$

where $f\{(x_1, y_1), \ldots, (x_{m+1}, y_{m+1})\}$ is the symmetric function

$$\frac{1}{m+1}[h\{(\|x_k - x_1\|, \|y_k - y_1\|), k = 2, \ldots, m+1\} +$$
$$\ldots + h\{(\|x_k - x_{m+1}\|, \|y_k - y_{m+1}\|), k = 1, \ldots, m\}].$$

## B   Additional Experiments

**Simulation 3: distributions of high dimensions.** In order to examine the effect of increasing the vector dimensions on the power, we sampled 100 observations from each of the following two distributions of a random vector of dimension $q \in \{10, 100, 1000\}$: $F_1 = \frac{1}{2}N_q(0, diag(1, \ldots, 1, 9)) + \frac{1}{2}N_q(0, diag(100, \ldots, 100))$ and $F_2 = \frac{1}{2}N_q(0, diag(9, 1, \ldots, 1)) + \frac{1}{2}N_q(0, diag(100, \ldots, 100))$. Therefore, the two distributions differ only in 2 coordinates out of $q$. Figure 1 shows the power of our novel tests. The choice of center points and univariate tests clearly affect the power. For this example, choosing an outlying value at the coordinate where there is a difference had the greatest benefit, and since this center point is tailored to where the distributions differ, the power was close to one for all $q$ using the $minP$ univariate test. For a single center point that is less optimal, or using all sample points as center points, the power deteriorated with the dimension $q$, and there was little power left at $q = 100$. Using $minP$ (detailed in § C) as the univariate test statistic resulted in greater power than the test of Kolmogorov-Smirnov (5), referred to as KS or the test of the Anderson and Darling family, constructed by (8) for the univariate two-sample problem, referred to as AD. Our competitors, Hotelling, Edist, and MMD had power at most 0.09 for all values of $q$ examined.

**Simulation 4: a closer inspection of relationship between power and sample size.** In order to examine the power increase as a function of sample size, we sampled $N \in \{25, 50, \ldots, 150\}$ observations from each distribution for several data generations. Figure 2 shows the power of our novel test using $minP$ as the univariate statistic, as well as that of Hotelling, Edist, and MMD. In the top panel, $F_1$ is a standard multivariate normal of dimension $q = 5$, and $F_2$ with coordinates each from the t-distribution with 5 degrees of freedom. The tests based on using $minP$ as the univariate test reach power close to one with $N = 150$, and MMD is a close second. Edist increases much slower, whereas Hotelling has no power to detect this alternative. In the bottom panel, $F_1 = \frac{1}{2}N_2(0, diag(1, 9)) + \frac{1}{2}N_2(0, diag(100, 100))$ and $F_2 = \frac{1}{2}N_2(0, diag(9, 1)) + \frac{1}{2}N_2(0, diag(100, 100))$. The tests based on using $minP$ as the univariate test have the highest power. As expected, the ideal single center point $z = c(0, 100)$ is the most powerful, but interestingly as the sample size increases taking all sample points as center points results in even greater power. The rate of increase when the center point is a single random sample point is far slower, as well as the rate of MMD and Edist. Hotelling has no power to detect this alternative.

**Real Data from the American Gut Project (AGP)**, preprocessed as described in (4). Data were analyzed for 1879 AGP participants. Unweighted UniFrac distance matrices were derived from the QIIME pipeline (4). We tested the null hyphotesis that the microbiome composition is the same for

Figure 1: The fraction of rejections at the $0.1$ significance level as a function of the vector dimension $q$ for testing $H_0 : F_1 = F_2$ when $F_1 = \frac{1}{2}N_q(0, diag(1, \ldots, 1, 9)) + \frac{1}{2}N_q(0, diag(100, \ldots, 100))$ and $F_2 = \frac{1}{2}N_q(0, diag(9, 1, \ldots, 1, 9)) + \frac{1}{2}N_q(0, diag(100, \ldots, 100))$, based on a sample of 100 points from each group, using different univariate tests (minP, KS, AD), and different center points schemes (all coordinates are 0 except for a value of 100 or 4 in a coordinate where the distributions differ, as well as using all sample points as center points). Based on 500 repetitions. The competitors Hotelling, Edist, and MMD had power at most 0.09.

males and females. Using the AD univariate test on the distances, where 256 sample points were selected at random to serve as center points, the $p$-value was $< 10^{-14}$ using both the Bonferroni and the Hommel combining method. This took $17.2 sec$ on a standard PC. For comparison, the Edist permutation test was run with 50000 permutations, and it took almost 4 minutes to run. It produced the smallest possible permutation $p$-value, which was $2 \times 10^{-5}$. To produce $p$-values in the order of $10^{-14}$ using a permutation test will be computationally impossible on a standard PC.

In practice, if the alternative is believed to be of simple form, one can apply a univariate test targeted to such an alternative (but which is not omnibus consistent). We tested the null hypothesis that the microbiome composition is independent of age using Spearman's test as the univariate test, and here again using 256 sample points selected at random to serve as center points, the $p$-value was $< 10^{-14}$ using both the Bonferroni and the Hommel combining method, and this computation took less than 20 seconds to run. For comparison, the $p$-value from the permutation test of (9), with 50000 permutations, was $8 \times 10^{-5}$, and it took 4 minutes to run.

## C  Description of the univariate tests in our experiments

In our simulations we utilized, as univariate tests, classical tests and also three modern tests proposed in (2). In this section we give a short description of these modern consistent tests.

We assume that $Y$ is a univariate continuous random variable, and that $X$ is categorical with $K \geq 2$ categories. We have $N$ independent realizations $(x_1, y_1), \ldots, (x_N, y_N)$ from the joint distribution of $X$ and $Y$. The test statistics only depend on the marginal ranks, and therefore are distribution free, i.e., their null distributions are free of the (unknown) marginal distributions $F_X$ and $F_Y$. The classical

Figure 2: The fraction of rejections at the 0.1 significance level as a function of sample size for testing $H_0 : F_1 = F_2$. Top panel: $F_1 = N_5(0, diag(1, \ldots, 1))$ and $F_2 = (t_5, \ldots, t_5 \ldots)$, we use minP as the univariate test for our novel test, and three different center points schemes (the center of mass, a randomly selected sample point, as well as using all sample points as center points). Bottom panel: $F_1 = \frac{1}{2}N_2(0, diag(1, 9)) + \frac{1}{2}N_2(0, diag(100, 100))$ and $F_2 = \frac{1}{2}N_2(0, diag(9, 1)) + \frac{1}{2}N_2(0, diag(100, 100))$, we use minP as the univariate test for our novel test, and three different center points schemes ($z = c(0, 100)$, a randomly selected sample point, as well as using all sample points as center points). Based on 500 repetitions. The power of the competitors Hotelling, Edist, and MMD are also shown.

Kolmogorov-Smirnov, Anderson-Darling and Cramer- von-Mises tests look at all possible partitions of the $y_i$'s into two parts and give a score to each partition, they then aggregate all these scores by either maximization or summation. On the other hand, the scores suggested in (2) aggregate scores for **all** possible partitions of the data, not just into two parts, and surprisingly this can be done in only $O(N^2)$ even though the number of partitions is exponentially large. Formally, for $N$ observations, there are $\binom{N+1}{2}$ possible cells, and $\binom{N-1}{m-1}$ possible partitions of the observations into $m$ cells, where a cell is an interval on the real line. Since the cell membership of observations is the same regardless of whether the partition is defined on the original observations or on the ranked observations, and the statistics only depend on these cell memberships, we describe the proposed test statistics on the ranked observations, $rank(Y) \in \{1, ..., N\}$. Let $\Pi_m$ denote the set of partitions into $m$ cells. For any fixed partition $\mathcal{I} = \{i_1, \ldots, i_{m-1}\} \subset \{1.5, \ldots, N - 0.5\}$, $i_1 < i_2 < \ldots < i_{m-1}$, $\mathcal{C}(\mathcal{I})$ is the set of $m$ cells defined by the partition. For a cell $C \in \mathcal{C}(\mathcal{I})$, let $o_C(g)$ and $e_C(g)$ be the observed and expected counts inside the cell for distribution $g \in \{1, \ldots, K\}$, respectively. The expected count $e_C(g)$ is the width of cell $C$ based on ranks multiplied by $N_g/N$, where $N_g$ is the total number observations from distribution $g$: $e_{[i_l, i_{l+1}]}(g) = (i_{l+1} - i_l) \times N_g/N$, where $l \in \{0, \ldots, m - 1\}$, $i_0 = 0.5$ and $i_m = N + 0.5$. For a given cell $C$, (2) considered Pearson's score or the likelihood ratio score:

$$t_C \in \left\{ \sum_{g=1}^{K} \frac{[o_C(g) - e_C(g)]^2}{e_C(g)}, \ \sum_{g=1}^{K} o_C(g) \log \frac{o_C(g)}{e_C(g)} \right\}. \tag{4}$$

For a given partition $\mathcal{I}$, the score is $T^{\mathcal{I}} = \sum_{C \in \mathcal{C}(\mathcal{I})} t_C$. In our experiments we used $t_C = \sum_{g=1}^{K} o_C(g) \log \frac{o_C(g)}{e_C(g)}$, so $T^{\mathcal{I}}$ is the likelihood ratio test statistic. The per partition test statistics can be aggregated over all partitions by summation (Cramer–von Mises-type statistics) or by maximization (Kolmogorov–Smirnov-type statistics):

$$S_m = \sum_{\mathcal{I} \in \Pi_m} T^{\mathcal{I}}, \quad M_m = \max_{\mathcal{I} \in \Pi_m} T^{\mathcal{I}}. \tag{5}$$

Tables of critical values for given sample sizes $N_1, \ldots, N_K$ can be obtained for (very) small sample sizes by generating all possible $N!/(\Pi_{g=1}^{K} N_g!)$ reassignments of ranks $\{1, \ldots, N\}$ to $K$ groups of sizes $N_1, \ldots, N_K$ and computing the test statistic for each reassignment. The $p$-value is the fraction of reassignments for which the computed test statistics are at least as large as observed. When the number of possible reassignments is large, the null tables are obtained by large scale Monte Carlo simulations. For each of the $B$ reassignment selected at random from all possible reassignments, the test statistic is computed. Clearly, the $B$ computations do not depend on the data, hence the tests based on these statistics are distribution free. Again, the $p$-value is the fraction of reassignments for which the computed test statistics are at least as large as the one observed, but here the fraction is computed out of the $B + 1$ assignments that include the $B$ reassignments selected at random and the one observed assignment, see Chapter 15 in (6). Finally, (2) also suggest another score which they call $minP$ that combines the $p$-values from each $m$, so that the test statistic becomes the combined $p$-value. Let $p_m$ be the $p$-value from a test statistic based on partition size $m$, be it $S_m$ or $M_m$. They consider as a test statistic the minimum $p$-value, $\min_{m \in \{2, \ldots, m_{\max}\}} p_m$ (where $m_{max}$ is typically $m/c$ for some constant $c > 1$. This combined $p$-value is a test statistic (not a $p$-value), and its null distribution can be easily obtained from the null distributions of the test statistics for fixed $m$s. Again, somewhat surprisingly, this test statistic can be calculated with the same effort as calculating the statistic for any fixed $m$ (i.e., $O(N^2)$). In our experiments, the $minP$ univariate test statistic is $\min_{m \in \{2, \ldots, m_{\max}\}} p_m$ where $p_m$ is the $p$-value of $S_m$.

## D  Technical definitions

**U-statistics**: For our purpose it suffices to discuss bivariate U-statistics based on i.i.d samples $x_1, \ldots, x_N$ and $y_1, \ldots, y_N$. Let $m$ be a positive integer and let $h((x_1, y_1), \ldots, (x_m, y_m))$ be a symmetric function. Let $\Pi_m$ denote the set of all subsets of size $m$ of $\{(x_1, y_1), \ldots, (x_N, y_N)\}$. Then $U_N = \sum_{S \in \Pi_m} h(S)$ is called a U-statistic and $h$ is called the kernel function. The theory of U-statistics was introduced by Hoeffding in (3) and has developed into a rich and useful method with many applications in asymptotic theory.

**Radon measure**: A measure $\mu$ is called inner regular, if, for any Borel set $B$, $\mu(B)$ is the supremum of $\mu(K)$ over all compact subsets $K$ of $B$.

A measure $\mu$ is called locally finite if every point has a neighborhood $U$ for which $\mu(U)$ is finite.

A measure $\mu$ is called a Radon measure if it is inner regular and locally finite.