[Reviews · NeurIPS 2016]

Reviewer 1

Summary

The paper introduces a two step procedure to test for association between two multivariate variables. The first step is to compute the distance between each sample and an arbitrary point, the second step is to apply a univariate test to the set of distances. It is shown that the null hypothesis on the set of distances is a necessary condition for the multivariate null hypothesis to hold, and therefore that the two step procedure using a consistent test for the univariate hypothesis is consistent for the multivariate problem. The use of several tests based on distances to several arbitrary points is also discussed.

Qualitative Assessment

The proposed approach and proofs of its properties are technically sound and novel to the best of my knowledge. The paper is clear and well written. I think this work could be a significant contribution to multivariate hypothesis testing but a few points would need to be fixed. I am mostly worried by the experimental section. I may have misunderstood something from table 1, but I don't understand why a single number (column) is given for the false positive rate (FPR) when four different scenarios are being tested. Is it pooled across the different scenarios? Wouldn't it make more sense to have, for each scenario, the FPR and TPR of each compared method? It could also be interesting to display ROC curves for a few methods. The introduced procedure also bears some similarity to tests based on random projection: the proposed method also amounts to building a lower dimension representation of the data (distance of each point to one or several centers) and to perform hypothesis testing in this new representation. Random projections are mentioned in the current manuscript but I think it would be important to discuss the relationship between the two approaches, in particular when the centers are picked at random, and to include random projections in the comparison. Finally it could be useful to clarify the gain expected from the transformation. Speed comes to mind, and is mentioned in the manuscript but not measured in the simulations. Gain in power (against multivariate testing in the original p-dimension space) is expected when p/n is not small enough. Is it possible to quantify this gain as a function of p/n either theoretically or empirically?

Confidence in this Review

2-Confident (read it all; understood it all reasonably well)


Reviewer 2

Summary

The paper presents a new framework for testing independence for multivariate random variables. The main idea is to compute the distance from a certain point (or set of points), and to use known univariate tests for independence on these distances. While various methods based on distances between points have been proposed before in the context of testing independence, the authors prove that if the univariate test statistic is consistent, then for all points except a measure zero chosen as center points, the multivariate test is also consistent. Moreover, if the univariate tests is distribution-free, then the multivariate test when using one center point, or for particular cases of using multiple center points, is also distribution-free, which can enable fast computation of a p-value for the test, without the need for a computationally costly permutation test.

Qualitative Assessment

The main contribution is proposing the first (as far as I know) consistent and distribution free test for the multivariate case (there have been several known such tests for the univariate case). This contribution is significant in the field of general non-parametric testing of independence, and the paper is overall well written and easy to follow. Therefore, in my opinion the paper is novel has a high potential impact, and clear, and thus worthy of publication. The technical level is adequate but not terribly impressive, as the idea presented by the author is quite simple and the proofs are short and mainly rely on results from Ref. [22]. Some comments below: The authors keep using 'consistent' for the univariate test, and 'power increasing to one' for the multivariate test. This distinction is unclear to me and seems confusing. Isn't the definition of 'consistent' being that power approaches one as sample size goes infinity when the alternative is true? (the definition is never given). Perhaps the issue is that the multivariate test is consistent only for almost all points, but not consistent for a subset of measure zero. Then one could still phrase this in terms of consistency, (say test is consistent except for measure zero points, or for example choose the center point randomly according to some continuous distribution) The authors prove in Theorem 2.2. that the multivariate independence test has power increasing to one for all z except for a set of measure zero. But this is true for every fixed point z. In practice, the authors propose using the tests in a data dependent manner (for example letting z be the average of the sample points, or letting z enumerate all data points). Therefore, as sample size grows z changes and it is not clear that such choices will not for example converge to 'bad' z values and give inconsistent tests. For example, in the example shown in Figure 1, if one chooses z as the average of the data points, then z will converge a.s. to the mean of the distribution (0,0) and then the test will have low power. This may be just a technicality, but needs to be addressed to justify the usage of the test with data-dependent z In page 7 "close inspection of a specific example" the authors suggest that if one has certain knowledge about the expected alternative, this could guide the choice of test statistics and center points, to improve power. But if such knowledge is present, then probably other methods could also work well - for example, you can perform the generalized likelihood-ratio test in case one has a parametric family. It would be good to compare the author's tests to such informative tests in terms of power. Another minor point with these simulations: The tests M_k and S_k are not formally defined. What is 'the optimal partition to five parts'? among which partitions? all possible real cutoffs, or as determined by the data? and what is 'optimal' for M_k? yielding most extreme chi-square test statistic? and S_k "sums up all scores for partitions into exactly k parts". What scores? what is the set of all partitions? (should be exponentially large). How is their sum computed? The authors claim that a certain multivariate test statistic is a U-statistic if the univariate statistic is a U-statistic. There is however no definition of U-statistic, explanations, arguments for why is it good to be a U-statistic (except a couple of sentences on asymptotics in the discussion), what does the order m of the statistic mean etc. Some more explanations are needed. The application to microbiome demonstrates a statistical and computational advantage for the author's test, but is not terribly interesting (there are far easier ways to distinguish between females and males). It would have been better to choose other classes or datasets (e.g. differences between healthy/disease, or disease stages or subclasses etc.) for which the data at hand together with improved independence tests, enable getting novel findings. The results on complex Radon measures of Ref. [22] are used in the main theorems of the authors. It would be good to explain what is a complex Radon measure, and how is it related to probability measures (as this is what is used in the proof in the Appendix). Also the notation |mu|(B) in page 2 is not explained (though guessable as absolute value of the measure).

Confidence in this Review

3-Expert (read the paper in detail, know the area, quite certain of my opinion)


Reviewer 3

Summary

The authors propose a modular framework for designing omnibus tests of independence between multivariate distributions. Given (X_i,Y_i) pairs, the authors propose designating a center point z_x and z_y for each, reducing to a bivariate problem by computing (||X_i-z_x||, ||Y_i-z_y||) for each i, and testing whether the resulting distances are independent using some off-the-shelf test of independence for the reduced problem. A similar construction is given for reducing to a univariate K-sample test if X is categorical with K categories. More generally, we can repeat the same process for several different choices of central points and adjust appropriately for multiplicity. This approach is appealing both for its generality and its computational simplicity, since using off-the-shelf bivariate independence tests lets us look up critical values in a table rather than having to compute them by, e.g., permutations. The authors show under mild conditions that their tests are consistent for almost every choice of central points z_x and z_y. They evaluate their methods in simulation, showing favorable power as compared to competing approaches.

Qualitative Assessment

The paper is well-written and well-structured. The organization is clear in the introduction and executed well throughout. The statements of the theorems are clear and there are few typos. The technical level is solid. The theorems are interesting, though I would have liked to see more refined results. I am by no means an expert in this area but your theorems did not, in my mind, establish that anyone would want to use the test. Power --> 1 against a fixed alternative, with an infinite sample, is a very low bar that would be cleared by many truly terrible tests (to name a terrible idea off the top of my head: Pearson's chi^2 test after discretizing the data into a multidimensional grid, where the discretization gets finer and finer as the sample size grows). The paper gave a nice, general framework encompassing a plethora of tests we could try but stopped short of giving specific recommendations, citing the need for further empirical and theoretical work to determine what tests will work well for various data types and against various alternatives. But one has certain alternatives in mind (as I suspect one would in most scientific applications) then why use an omnibus test? My biggest reservation has to do with practical utility, which is, in my view, questionable. In what concrete applications will your approach actually represent an improvement on already-available methods? And what do we really learn when an omnibus test of independence rejects the null? Formally rejecting independence, unless we can make a concrete statement about how the distributions differ, doesn't seem very useful -- for example, was there ever any doubt whether men's and women's microbiomes were identically distributed, and what do we learn by declaring that they are not? All but one of the simulation scenarios were for two-sample tests in dimension 2. The other example was for dimension 5, but there you were comparing a very heavy-tailed distribution to a very light-tailed one, so it is unsurprising that a procedure based on the distribution of inter-point distances, or distance from the centroid, would do well. These are very low-dimensional examples, leaving the reader to wonder whether your tests will have any appreciable power in intermediate or high dimensions. On this point, I did not feel that the AGP experiment added anything to the paper; it sounds like the null was very obviously not true, and every test was able to pick that up very easily. It would be more interesting to compare the power of different approaches by subsampling the data. Bottom line: there are several valid ways to make the case for your method over already-existing methods: rigorous theory, heuristic arguments, simulations, or on a real data set that represents an important use case. But in my judgement you haven't succeeded on any one of these counts. Your theoretical results can't make the case for your method over other consistent methods; you don't really give heuristic arguments about why or when your method should be more powerful; your simulations are unconvincing because they don't really go beyond dimension 2, and in your real data experiment all methods apparently have good power. Minor comments: Table 1: The numbers in this table should be reported with most two significant digits. For example, 250 rejections on 500 tries should give estimate 0.5 and standard error .022. 100 rejections on 1000 tries in the first column gives estimate 0.1 and standard error 0.013. More generally, please report standard errors (if only by reporting a maximum standard error of 0.022 for all values given in the table). Prop 2.1: this is stated much more generally than what you seem to actually need; taking the time to parse the definitions involved briefly derailed my otherwise-smooth progress through the paper. Can you state it in a simpler way? - Please state somewhere that you mean Euclidean distance (I assume you do) - L.77: R^p formatted differently than elsewhere - Labels on Fig. 1 are very small - You define the term "distribution-free" as "the null distribution of the test statistic does not depend on the marginal distributions of X and Y." In your proposal to combine p-values via e.g. Bonferroni, the above definition would not apply: the minimum p-value's null distribution does, in fact, depend on the dependence of the p-values (of course, the test still works). Please resolve this issue, e.g. by modifying your definition to say "the critical value of the test statistic ..." - L.150: "two sample" -> "two-sample" - L.219: the multivariate t-distribution is usually defined not as a product of univariate t-distributions, but rather as a multivariate normal divided by an independent chi-squared distribution. I would call this a "product t-distribution" or something. - L.281: "hypohtesis"

Confidence in this Review

2-Confident (read it all; understood it all reasonably well)


Reviewer 4

Summary

The paper defines a general class of methods for testing independence between two random vectors. Each sample is summarised into a vector of distances to one or more focal points. This new univariate distribution of distances characterises the multivariate distribution, so tests of independence for one-dimensional distributions can be used. It is proven that for almost focal point, the univariate distance distributions would not be independent if the source distributions were not independent; therefore consistency of the test depends only on the consistency of the univariate test. A similar reduction can be used testing equality of K-samples. The class of methods generalises several very recent contributions to multivariate non-parametric independence tests.

Qualitative Assessment

The paper is well written and presents well a generalised framework for comparing multivariate samples. It gives an easy to follow prescription for generating new tests for independence or equality of distributions. In particular, the result in Theorem 2.2. that multivariate differences would result in univariate distances from a focal point, for almost any point, is an interesting and attractive results. The experimentation is also nice. The reason the marks are not higher are that this paper generalises ideas already found in previous papers from recent years, in particular [11-12]. In that sense, this is not quite a breakthrough but rather an expansion of previous ideas. Nevertheless, these ideas open new channels for future investigation. Technically, I would also like to note that Theorem 2.2, although very attractive, does not necessarily apply to the cases practically used later in the experiments. If distances are measured to any arbitrary fixed point, then the Theoremm as stated suffices. But if, instead, distances are measured to an observed "center of gravity", we could imagine the focal points converging as sample size increases to a particular limit; then this limit might be exactly the zero-measure point under which the two distributions are similar.

Confidence in this Review

2-Confident (read it all; understood it all reasonably well)


Reviewer 5

Summary

This paper suggests a new method for multivariate independence testing. The idea is to pick, deterministically or randomly, one or a number of center points. Compute the distances from sample points to the selected center points and do univariate independence testing between those distances. Theorems proved in the paper show that except for possibly a set of Lebesgue measure 0, for all center points, the univariate distance independence tests yield the multivariate test. This nice result hinges on the fact that the Euclidean distance has at most exponential-quadratic growth.

Qualitative Assessment

The results are interesting and promising. This paper could benefit from a bit more experimental results for instance to shed light on the running time, empirical power, pitfalls, etc. in comparison to other techniques. Also, a discussion on different distance metrics could be very enlightening. This paper proves the results for the Euclidean 2-norm. For what types of metric spaces and what other types of metrics do the results still hold? For instance, discrete topological/metric spaces? infinity-norm? 1-norm? Hilbert spaces? The paper starts with the sentence "Let ..." which is the math/theory style of presentation. This reviewer loves that style and considers self a mathematician/theoretician, but for the NIPS community that might be a bit out-of-standard.

Confidence in this Review

2-Confident (read it all; understood it all reasonably well)


Reviewer 6

Summary

This paper proposed a way to test null hypothesis of multi-dimensional vectors, by using uni-variate tests via distance from arbitrary points. The authors provided some theoretical background behind the methods. Some experimental data showed reasonable performance while the computational complexity can be significantly reduced compared to existing methods.

Qualitative Assessment

The way to show the advantage of the method via experimental data should be re-considered because the readers cannot easily assess the method in terms of the power and complexity compared to existing approaches. In addition, the achieved gain of computational complexity is rather little less than a magnitude, which limits the usefulness and impact of the method. It should be re-organized to emphasize the contributions.

Confidence in this Review

1-Less confident (might not have understood significant parts)